# Role of dams in reducing global flood exposure under climate change

Julien Boulange [1]✉, Naota Hanasaki [1], Dai Yamazaki [2] & Yadu Pokhrel [3]

Globally, flood risk is projected to increase in the future due to climate change and population growth. Here, we quantify the role of dams in flood mitigation, previously unaccounted for in global flood studies, by simulating the floodplain dynamics and flow regulation by dams. We show that, ignoring flow regulation by dams, the average number of people exposed to flooding below dams amount to 9.1 and 15.3 million per year, by the end of the 21st century (holding population constant), for the representative concentration pathway (RCP) 2.6 and 6.0, respectively. Accounting for dams reduces the number of people exposed to floods by 20.6 and 12.9% (for RCP2.6 and RCP6.0, respectively). While environmental problems caused by dams warrant further investigations, our results indicate that consideration of dams significantly affect the estimation of future population exposure to flood, emphasizing the need to integrate them in model-based impact analysis of climate change.

[1] National Institute for Environmental Studies (NIES), Tsukuba, Japan. [2] Institute of Industrial Science, The University of Tokyo, Tokyo, Japan. [3] Department of Civil and Environmental Engineering, Michigan State University, East Lansing, MI, USA. ✉email: boulange.julien@nies.go.jp

Global warming is expected to increase flood risk by altering the distribution, variability, and intensity of precipitation events[1,2]. While global estimates of populations exposed to river flooding vary widely across studies, a 4–20 fold increase by the end of the 21st century is commonly predicted[3–5]. To mitigate the destructive potential of floods and maximize water availability for human consumption, an estimated 2.8 million dams[6] have been constructed globally with a total water impoundment capacity ranging from 7,000 to 10,000 km³, which represents over one-sixth of the annual continental discharge to global oceans[7–9]. Currently, about half of the planet's major river systems are regulated by dams[10,11] and only 23% of rivers worldwide flow uninterrupted to the ocean[6]. By regulating water flow, dams generally alter the frequency, duration, and timing of annual flooding events[12]. With more than 3,700 major dams planned or under construction worldwide[13], understanding the role of dams in climate impact studies has become increasingly important. Previous studies on flood prediction, however, have neglected the role of dams[3,14] due to data scarcity[15], difficulties in parameterizing reservoir outflows, and challenges in implementing features of dams that function at a scale smaller than those accounted for by global-scale models.

Previous global-scale analyses of floods have reconstructed historical flood patterns[16,17] to forecast future floods considering climate change[3,14] and/or socio-economic development factors[18,19]. A key conclusion of the Fifth Assessment Report (AR5) of the Intergovernmental Panel on Climate Change (IPCC) was that the number of people exposed annually to the equivalent of a historical 100-year river flood was projected to triple when compared to high and low emission scenarios. However, despite the regulation of most large rivers by dams, the extent to which their alterations of river and floodplain dynamics interacts with flooding, and the exposure of populations to floods in response to climate change remains largely unknown since dams have not been physically integrated into global flood-impact studies[3,14,15,20]. The few studies that have accounted for dams and/or flood protection have underscored the importance of considering dam-induced changes in streamflow characteristics in flood-hazard modelling[21–23]. In the contiguous United States (CONUS), dams are reported to reduce total flood exposure by 9% (protecting approximately 590 million people) owing to the medium to high dam attenuation effects on the 100-year return period discharge of 62% of CONUS hydrological units[22].

Here, we provide the first global assessment of the role of dams in reducing future flood risk under climate change by using a modelling framework that integrates state-of-the-art global hydrological model with a new generation of global hydrodynamics model. Specifically, the modelling framework quantifies changes in the frequency of historical 100-year return period floods when dams are considered and estimates the global population at a reduced risk of flood exposure. Throughout this study, a flooding event is defined as extreme discharge associated with a 100-year return period (probability). We specifically investigated flood frequency (number of floods per year), the associated maximum flooded area, and populations exposed to these floods.

## Results

**Streamflow regulation.** Robust and reliable estimates of future river floods rely on two critical components: accurate reproduction of river discharge and appropriate prediction of floodplain inundation dynamics. In this study, we used two different models to simulate these critical processes globally. River discharge considering dams was simulated by H08, while flood inundation dynamics were simulated by the CaMa-Flood model. H08 is a global hydrological model that considers human interactions with the hydrological cycle. CaMa-Flood is an advanced river hydrodynamics model with an emphasis on efficient flow computation at the global scale (see Methods). Two global flood simulations were performed: one considering dams and one not considering dams. In total, four bias-corrected global circulation models (GCMs) combined with three radiative forcing scenarios (historical, RCP2.6, and RCP6.0) were used to force the models (see Methods).

The H08 model has been widely used and validated in global studies and accurately reproduces monthly river discharge in basins heavily affected by anthropogenic activities[24]. At the global scale, the H08 model has been benchmarked against other global hydrological models (GHMs) and has performed relatively well for reproducing the magnitude of high flows associated with different return periods[25]. H08 has also been calibrated and validated at finer spatial and temporal resolutions in multiple regional analyses, including the Chao-Phraya basin, the Ganges–Brahmaputra–Meghna basin, and Kyushu Island, among others[26–28]. Critical to these faithful discharge reproductions is the scheme used for dam operations. While improvements to the dam operation scheme implemented in H08 have been recently proposed[29,30], it is still regarded as the benchmark to beat, given its ability to capture observed reservoir storage variation with high accuracy[31]. CaMa-Flood has also been extensively used and validated. It is capable of faithfully reproducing historical flood patterns[32–34] and daily measurements at river gauging stations across the globe[33], partly owing to the integration of satellite-based topography data[35]. While both models have been widely used for climate impact assessments, they have never been coupled to analyze global-scale floods, leaving a gap in our understanding of the potential role of dams in reducing future flood risks. While the GCMs employed in this analysis were not assimilated, and consequently do not reproduce the exact timing of historical weather events, we nevertheless confirmed that our coupling framework can satisfactorily reproduce observed monthly discharges before and after dam construction (see Supplementary Figs. 13–23) and that its predicted maximum discharges in 33 large basins were reasonably similar to available observations (see Supplementary Fig. 24). We further compared global patterns of future floods with a previous publication[3] (Supplementary Figs. 1 and 5a, b). We also compared the historical and predicted populations exposed to 100-year floods with information from published literature and a public database (see Supplementary Table 2 and Supplementary Note 1).

**Population exposure to floods.** Results indicate that, driven by climate change, the risk of floods will increase in the future. However, owing to the implementation of dams in our simulations, on average (range from the first and third quartiles in bracket represent uncertainty from the GCM ensemble), populations exposed to flooding below dams decreased by 16.3% (5.7–30.7%) in the RCP2.6 scenario and 12.8% (4.2–27.5%) in the RCP6.0 scenario, respectively, compared to the RCP simulations not considering dams (over 2006–2099, see Fig. 1). The decrease in the number of people exposed to floods due to the implementation of dams was highest during the last decade of the 21st century for both RCPs. On average, 9.1 (4.6–18.1) million people were exposed to river floods in RCP2.6 (no dams) compared to 7.2 (3.5–15.1) million people in the simulation with dams. In the RCP6.0 scenario, the population exposed to river floods increased considerably to 15.3 (8.3–27.2) million and 13.4 (7.3–24.3) million for the simulations without and with dams, respectively. Large differences, consistent across experiments, in the number of people exposed to floods between the GCMs were apparent

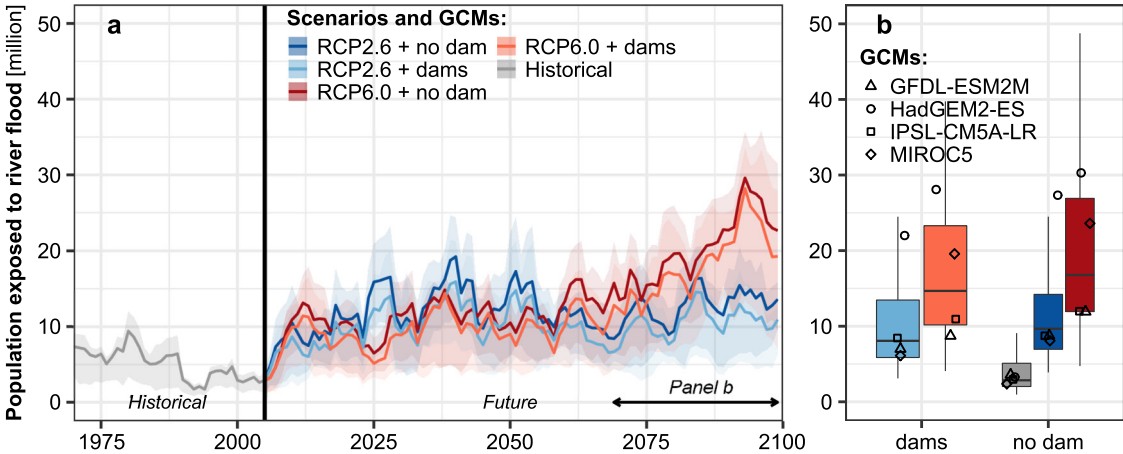

**Fig. 1 Population exposure to the historical 100-year river flood (keeping population constant at 2010 level) downstream of dams. a** 5-year moving averages of the population living below dams exposed to the historical 100-year river flood for historical (grey line) and future simulations for 2 RCPs and experiments (colour lines). The uncertainty range represent the spread among GCMs. **b** The 95th and 5th range (whiskers), median (horizontal lines in each bar), and 1st and 3rd quartiles (height of box) and individual mean values among GCMs (markers) of the population exposed to the historical 100-year flood for grid-cells located below dams over the 2070–2099 period.

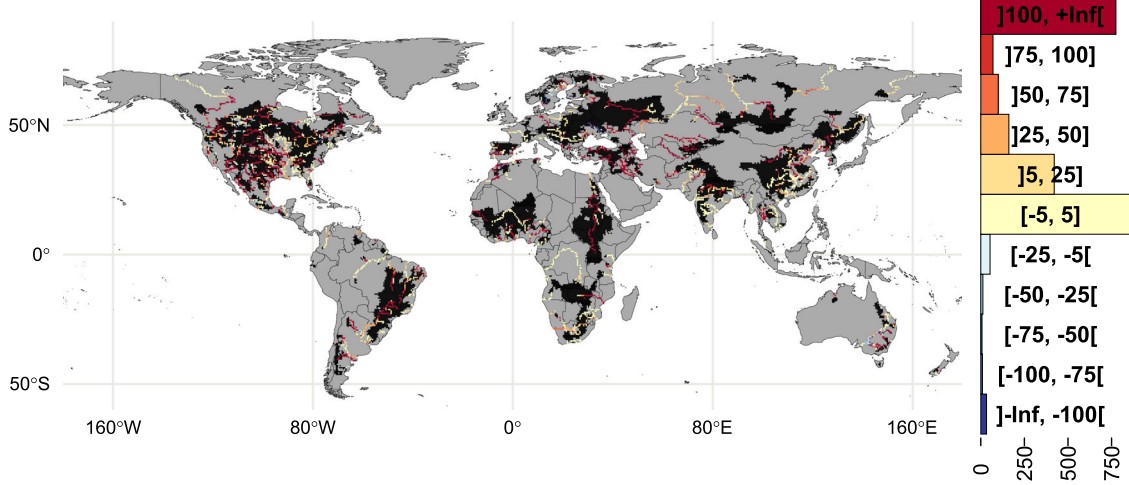

**Fig. 2 Projected difference in multi-model median return period (years) in 21 C for discharge corresponding to the historical 100-year flood between the experiment not considering dams minus the experiment considering dams.** Grid-cells belonging to Köppen–Geiger regions *BWk*, *BWh* (hot and cold desert climates, respectively), and *EF* (ice cap climate) and for which the 30-year return period discharge was lower than $5\,\mathrm{m\,s^{-1}}$ were systematically screened out (see Methods). The case for representative concentration pathway (RCP) 6.0 is shown (RCP2.6 available in Supplementary Fig. 5c).

(Fig. 1b). When population growth was taken into consideration using Shared Socioeconomic Pathways (SSPs) (see Methods), accounting for dams reduced populations exposed to flooding below dams by 20.6–32.0% for RCP2.6 and 7.0–16.8% for RCP6.0 (lowest and highest values across the five SSPs).

**Return period of future floods**. Downstream of dams, historical 100-year floods occurred less frequently in the experiment considering dams than in the experiment with no dams for: (on average and ± standard deviation across GCMs), 66.6 ± 4.2% and 60.8 ± 12.7% of the grid-cells in RCP2.6 and RCP6.0, respectively (Fig. 2, Supplementary Fig. 5c). These results are similar to other regional- and country-scale analyses. For example, in the US, medium or large dam-attenuation effects were reported for 62% of hydrologic units[22]. Likewise, a study in Canada revealed that dams totally prevented flows with a return period greater than the historical 10-year recurrence[36] (see additional comparison with existing studies in Supplementary Note 3). Particularly prominent

reductions in future flood frequency were observed along major sections of rivers containing multiple high-capacity dams (e.g. the Mississippi, Danube, and Paraná; see Supplementary Fig. 2). Reductions in 100-year flood frequencies in the experiments involving dams decreased moving downstream, becoming relatively small (or negligible) at the river mouth (e.g. in the Amazon, Congo, and Lena; see grey cells in Fig. 2). In a few locations (blue cells in Fig. 2), the presence of dams increased the frequency of historical 100-year floods compared to experiment without dams (6.7 ± 2.4% and 4.6 ± 1.1% for RCP2.6 and RCP6.0, respectively). This behaviour was connected to sporadic overflow events referred to as the pulsing effect by Masaki et al.[37] and has been documented for some rivers in the US[23]. Although water released from dams was regulated through the majority of the simulation period, pulsing events can result in a dam failure to prevent flooding, distorting the distributions of extreme discharge, and compromising the fitting of the extreme discharge to a Gumbel distribution (see Methods). In such cases, the definition of the 100-year flood is rather ambiguous, and while great efforts are

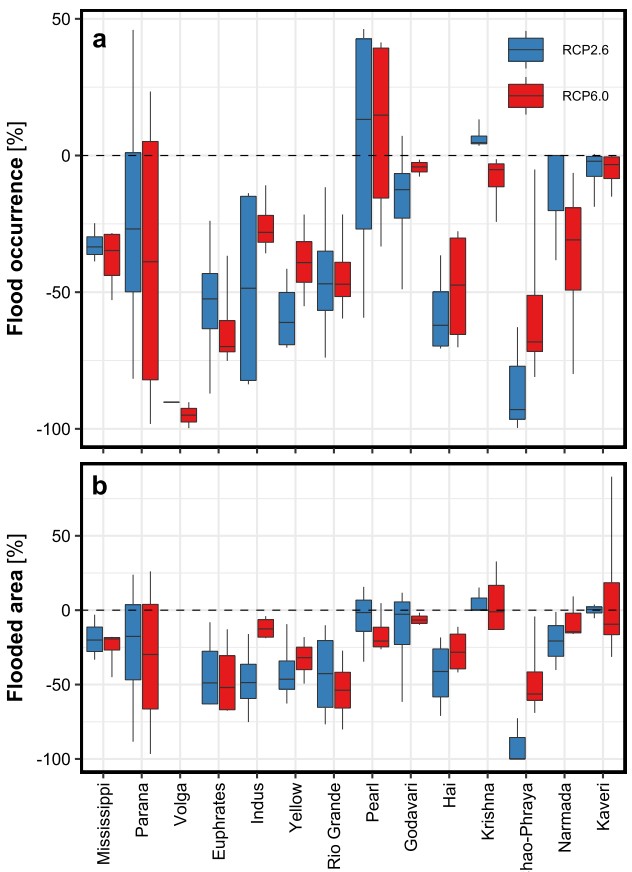

**Fig. 3 Changes in the occurrence of historically 100-year river flood and maximum flooded area in the late 21st century for 14 catchments resulting from simulations considering and not considering dams.**
**a** Occurrence of the historical 100-year river flood and, **b** annual maximum flooded area over the period 2070–2099, given two experiments (with and without dams), and tow representative concentration pathways (RCP). The box-and-whisker plots include the 95th and 5th range (whiskers), median (horizontal lines in each bar), and 1st and 3rd quartiles (height of box) of the annual values obtained for all four global circulation models.

made to prevent overflow[29], not all are reflected in the generic scheme for dam (see Methods). Note that since the lead time before major storms is generally too short for preventive dams emptying, pulsing may not be totally averted in global dam simulations.

**Evolution of future floods for individual catchments.** Median changes in the occurrence of historical 100-year river floods and the maximum flooded areas in the experiment considering dams relative to the experiment not considering dams were computed over the 2070–2099 period for 14 catchments (see Methods for the selection of catchments). Figure 3 indicates that the historical 100-year floods occurred less frequently in the experiment with dams, decreasing, on average, across catchments by 36.5% (26.6–49.1%) for RCP2.6 and 35.5% (28.8–46.6%) for RCP6.0. Similarly, the maximum flooded area in the catchments shrank on average by 22.5% (19.8–40.5%) and 25.9% (12.1–34.5%), for RCP2.6 and RCP6.0, respectively. These reductions in the occurrence of 100-year floods and maximum flooded areas were robust to the choice of extreme discharge indices used for iden- tifying flood events (see Methods), with the exception of two catchments that experienced pulsing from dams (Supplementary

Fig. 7). We note that by employing alternative extreme discharge indices (see Methods) to identify flood events, the eventual influence of pulsing events on the occurrence of 100-year floods and maximum flooded areas was largely filtered.

The 100-year return extreme discharge expected in the future (2070–2099) was calculated for all combinations of RCPs and experiments (Supplementary Fig. 8) along the main river of the 14 catchments. Downstream of dams, the experiment considering dams always produced a lower 100-year discharge than that produced by the experiment not considering dams. For catch- ments located in regions where annual precipitation and/or snowmelt is forecast to decrease in the future (the Mississippi, Volga, and Euphrates; see Supplementary Figs. 1 and 8a, c, d), the RCP2.6 simulations produced higher 100-year discharges than those in the RCP6.0. However, simulations employing the RCP6.0 scenario and the experiment not considering dams generally produced the highest 100-year discharges. For catch- ments containing few dams on the mainstem river, future 100- year return extreme discharges in both experiments (with and without dams) were similar at the river mouth (Supplementary Fig. 8i, k, l, m, n). However, in other catchments, the 100-year extreme discharges were clearly reduced in the experiments considering dams (Fig. 3 and Supplementary Fig. 7), resulting in reduced flood exposure to populations residing downstream of dams. In addition, the reductions in 100-year extreme discharge in the Amazon, Congo, and Mekong rivers were relatively small due to the small cumulative storage capacity of the mainstem dams compared to the discharge volume generated in these basins.

## Discussion
Explicitly considering dams in climate-impact studies of floods significantly offsets the population size exposed to river floods. Downstream of dams at the end of the 21st century, a 100-year flood was, on average, indicated to occur once every 107 (79–168) years for RCP2.6 and once every 79 years (55–103) in the experiments not considering dams (see Supplementary Fig. 8). In RCP6.0, the historical 100-year flood occurred more frequently: once every 59 years (39–110) and 46 years (33–75) for the experiments considering and not considering dams, respectively (see Supplementary Fig. 8). In most catchments, dams reduced both the frequency of floods and the extent of flooded areas. Our findings were robust to the selection of indices used to identify floods although the pulsing effect of dams was identified as compromising estimates in some catchments. This problem could be partially mitigated by revising the reservoir operation method used in the present study by accounting for future precipitation variabilities and cascade-dams. Since our large-scale modelling considers daily precipitation, potential dam failure due to increased extreme precipitation events[38] (resulting in down- stream flooding) is not fully considered here, nor are the con- struction and filling phases of a dam's life cycle. Nevertheless, neglecting the morphological, environmental, and societal impact of dams[39], our results imply that dams significantly decrease the risk of future global floods in terms of both frequency and intensity, protecting 1.4 (0.7–3.1) and 2.3 (0.8–3.7) million people at the end of the 21st century, for RCP2.6 and RCP6.0, respectively.

The aging dam landscape faces new temperature, snow, dis- charge, and floods patterns that increase the risk of hydrological failure[40,41]. To maintain historical levels of flood protection in the face of climate change, new dam release operations will be required. In addition, precise and reliable hydro-meteorological forecasts will be invaluable for maximizing flood protection and avoiding untimely and excessive outflows. By focusing solely on

the role of dams in reducing global flood exposure under climate change, the results of this study are perceived as over emphasizing the benefits of dams (see Supplementary Note 2). However, given the many negative environmental and social impacts of dams[39], comprehensive assessments that consider both potential benefits and adverse effects are necessary for the sustainable development of water resources. Furthermore, future analyses of global flood risks would benefit from: addressing the disparities and uncertainties associated with global dam and river datasets (e.g. location, characteristics, networks); developing realistic future population projections that account for population behaviour; enhancing historical GCM scenarios by assimilating past observations; and archiving and referencing historical reservoir operations, streamflow, and inundation for robust model validation.

## Methods

**Models**. Two hydrological models were used in this study. H08 is an open-source global hydrological model (GHM) that explicitly considers human water abstraction from six major water sources including dams[24]. The reservoir operation scheme in H08 is a generic one; that is, it is not tailored to a specific site. A detailed description can be found in Hanasaki et al.[31]. Outflow from dams is computed in two steps: considering the water currently available in the reservoir, a provisional annual total release is computed, and is then adjusted every month according to changes in storage, inflow, and water demand below the dams. The algorithm distinguishes two classes of dams: irrigation and non-irrigation dams, which influences the computation of monthly water release. It should be noted that, while the storage capacity used in the simulations corresponded to that reported in the Global reservoirs and Dams database (GRanD), the actual storage capacity of dams is expected to be lower due to the allocation of dead and surcharge storages. As a result, the allocated dam storage in the present simulations is likely to have been overestimated. The most recent version of the H08 model, which participated in ISIMIP2b, was employed[24]. Simulations were carried out at a spatial resolution of 0.5° by 0.5°, and a 1-day interval.

CaMa-Flood is a new generation of global river routing model that relies on HydroSHEDS[42] topography to simulate floodplain dynamics and backwater effects by explicitly solving the local inertia equation[33]. The model was reported to outperform other GHMs for reproducing historical discharge[43]. The CaMa-Flood model requires only daily runoff as an input, and by computing the inflow from upstream cells and outflow to downstream, the evolution of water storage can be predicted. In this study, three output variables were used: the total discharge exiting a grid-cell (sum of river discharge and floodplain flow), the flooded area, and the flooded fraction of a grid-cell. To output the latter two variables, CaMa-Flood assesses whether water currently stored in a grid-cell exceeds the total storage of the river section. When this is the case, excess water is then stored in the floodplain, for which topography (dictated by HydroSHEDS) controls the flood stage (water level and flooded area).

To simulate the effects of water regulation due to anthropogenic activities on floodplain dynamics, the H08 and CaMa-Flood models were coupled because, in its current global version (v3.62), the global version of CaMa-Flood cannot simulate dam operations despite being essential for assessing flood risk. Hence, the H08 model is required for accurate forecasts of dam outflow. To ensure compatibility between the models, the river network originally used in CaMa-Flood was employed in both models. The coupling procedure is as follows: simulations with the H08 model are conducted; the daily runoff predicted by H08 is used as a forcing input in CaMa-Flood; in grid-cells containing major dam(s),[44] the river discharge produced by H08 (following the reservoir operating rule) is imposed onto the CaMa-Flood model (Supplementary Fig. 3a); the difference in daily discharge between the two models due to water regulation is added to the hypothetical storage associated with every dam but without interacting with the river or floodplain to close the water balance.

For grid cells that are neither downstream nor upstream of dams (light blue locations in Supplementary Fig. 3), experiments considering and not considering dams produced the same discharge outputs. In contrast, for grid cells located below and above dams, the daily discharge simulated by the experiments considering dams can change compared to the experiments not considering dams due to water regulation (below dams) and the impossibility of the backwater effect and its propagation (above dams).

**Input data**. The four general circulation models (GCMs; GFDL-ESM2M, Had-GEM2-ES, IPSL-CM5A-LR, and MIROC5) implemented in the ISIMIP2b protocol participated in the Fifth Assessment Report (AR5) of the Intergovernmental Panel on Climate Change (IPCC). The forcing data consisted of precipitation, temperature, solar radiation (short and long wave downward), wind speed, specific humidity, and surface pressure which were bias corrected[45] and downscaled to a 0.5° by 0.5°-grid resolution. Here we used three radiative forcing scenarios:

historical climate (1861–2005), and two future scenarios consisting of a low greenhouse gas concentration emission scenario (RCP2.6; 2006–2099) and a medium–high greenhouse gas concentration (RCP6.0; 2006–2099). Note that the historical climate scenario does not attempt to reproduce the exact day-to-day historical climate but rather gives a consistent evolution of the climate under a given climatic forcing.

Dam specifications (location, storage capacity, and construction year) are provided in GRanD[44,46]. The dams were georeferenced to the river network employed in CaMa-Flood, iteratively adjusting dam locations when necessary until the catchment areas of each dam reported in GRanD corresponded to ± 10% of the catchment area in CaMa-Flood[47].

**Experiments**. For the future scenarios (RCP2.6 and RCP6.0), two experiments were considered. In the first experiment, dams were not implemented, therefore this simulation is analogous to the simulations conducted in previous studies[3,14]. In contrast, in the second experiment, the effect of major global dams on water regulation, hence floodplain dynamics, were considered. Due to water regulation, the future return period (in years) associated with the historical 100-year extreme discharge might change compared to that obtained for the experiment not considering dams (Supplementary Fig. 8). These potential differences were used to quantify the effect of dams on the potential reduction in the future return period of the historical 100-year flood.

**Validation**. The H08 model has been extensively validated in catchments located in India, the US, China, Europe, and South America for predicting river discharge, total water storage anomalies, groundwater, and water transfer[24]. Across these major catchments, the average Nash–Sutcliffe efficiency (NSE) obtained when comparing daily observed and simulated discharge was positive. Benchmarked against GHMs, H08 was reported to perform relatively well for reproducing historical daily discharges[25]. More relevant to the context of this study, the same study[25] highlighted that the H08 model was among the top four GHMs best able to reproduce the magnitude of extreme discharge and the maximum flows associated with different return periods.

The ability of the CaMa-Flood model to reproduce floodplain inundation was reported in the Amazon basin, where it performed well[33]. In addition, the discharges produced by CaMa-Flood have been evaluated against gauge observations in 30 major river basins[33]. CaMa-Flood has also been benchmarked against nine GHMs, including the H08 model, at 1701 gauge locations[43]. Generally, discharge simulations using CaMa-Flood produce lower and later peak discharges compared to those predicted by other GHMs, resulting in more accurate reproduction of observations[43].

The quality of discharge data produced by nine GHMs, including the H08 model used in this study, was evaluated and compared against calibrated regional hydrological models in 11 large river basins[48]. While regional models generally outperformed GHMs in most regions, GHMs reproduced the intra-annual variability of water discharge reasonably well. Extreme discharges are strongly related to floods,[5] and the inclusion of human activity in hydrological simulations, such as in H08 has been reported to greatly improve the reproduction of hydrological extremes[49]. The predicted return period for the historical 100-year discharge obtained in the experiment not considering dams was compared to the literature. Global estimates of populations exposed to river floods were also compared to those reported in the literature (Supplementary Table 2). We evaluated how the coupled model reproduced river discharges before and after the implementation of dams at key locations. We separated our observation dataset into two parts: pre- and post-dam construction. We then compared our dam and no-dam simulations to the relevant observations. Supplementary Table 3 lists the dam locations of the dams and their key characteristics.

**Definition of flood event and extreme discharge**. We compared the frequency of historical (1975–2004) and predicted future (RCP2.6 and RCP6.0; 2070–2099) flood events using given two experiments: an experiment in which no dams were considered (analogous to previous studies[3–5]), and an experiment considering global dams (Supplementary Fig. 2)[50]. Flood events were defined as the historical 100-year return extreme discharge, that is, the extreme discharge with a probability of exceeding 1/100 in any given year.

Two annual-extreme discharge indices were used in this analysis to assess the robustness of our findings expressed by the spread (or consistency) of results from multiple GCMs and extreme indices. We primarily focused on the maximum annual daily discharge ($P_{max}$) since it is the preferred index used in the literature[3–5,14]. The alternative indicator is the annual 5th percentile ($P_{05}$) of daily discharge.

Before fitting the Gumbel distribution to estimate the 100-year river discharge, we initially compared the two series of extreme discharges in the dam and no-dam experiments. Run-of-the-river dams tend to alter the natural flow regimes only negligibly. For such locations, the fitted Gumbel distribution should be identical in both experiments. In contrast, in rivers heavily regulated by dams, it is possible that the extreme discharge series obtained for the experiment considering dams included many identical or tied values. We initially computed the absolute difference between the annual discharge extremes obtained by the simulation not considering dams minus the simulation considering dams and compared that

difference to a given threshold (150 m³ s⁻¹, or an annual difference of 5 m³ s⁻¹ between the extreme discharge generated for the experiments with and without dams). When the threshold was exceeded, the extreme discharge series were considered dissimilar and therefore treated separately. In contrast, when the threshold was not exceeded, the two extreme discharge series were considered similar and all data were pooled before moving to the fitting phase. We assessed the sensitivity of our results to alternative thresholds, with those results reported in Supplementary Table 1.

**Fitting of Gumbel distribution**. The extreme discharges were first ranked in ascending order and fitted to a Gumbel distribution using the L-moment method[51]. As a result of the comparison protocol, the number of data to fit was either 60 (experiments with and without dams produced similar extreme discharges and were pooled) or 30 (experiments with and without dams produced different extreme discharges). The fitting process is identical to that described in detail in the Supplementary Note 2 of Hirabayashi et al.[3].

**Assessment of goodness of fit**. The goodness of fit of the annual extreme discharge to the Gumbel distribution was assessed using the probability plot correlation coefficient test (PPCC)[52]. While other methods can be used to assess the goodness of fit of the Gumbel distribution, the PPCC has been reported to outperform most of them in terms of rejection performance[53]. The PPCCs were computed for all historical simulations and are reported in Supplementary Fig. 9. A PPCC score close to 1 indicates that the distribution of the extreme series is well fitted by the Gumbel distribution. For a sample size of 30, the critical PPCC score at the 95th level of significance was reported[52] to be approximately 0.96.

**Bootstrap**. A bootstrap methodology was used to assess the influence of the 30-year samples on the fitted Gumbel distribution[54]. We generated 1000 bootstrap estimates for every GCM and all experiments. We did not explore all combinations of bootstrap estimates and GCMs due to the high computational cost (1012 estimates for a given year and a single experiment). Instead, we ranked the estimates in descending order before taking the average across GCMs (1000 estimates for a given year and a single experiment). While simple, this method has the advantage of reporting the broadest confidence intervals since the lowest and highest estimates among GCMs are averaged.

**Masking**. In the reported global maps, we masked grid-cells belonging to the Köppen–Geiger regions BWk (hot desert climates), BWh (cold desert climates), and EF (ice cap climates) which discharge corresponding to the historical 30-year return period was less than 5 m³ s⁻¹ (Supplementary Fig. 4). In such grid cells, flooding is not a problem due to the low volume of water discharge. As a result, the goodness of fit of the Gumbel distributions was generally low (as indicated by a low PPCC score in Supplementary Fig. 9).

**Population exposure**. The population dataset, created by the Socioeconomic Data and Applications Center (SEDAC), consists of the Gridded Population of the World (GPW, v4.11) for the year 2010[55]. The population was fixed at 2010 to assess only the effect of climate change on population exposure to floods. To increase the accuracy of our exposure assessment, the original 0.5° resolution flooding depths were downscaled to a resolution of 0.005°. The file containing flooding depth resulting from historical 100-year floods was constructed annually following a two-step procedure. First, we determined the 0.5° grid cells experiencing a 100-year flood as indicated by the annual discharge extreme exceeding the 100-year historical discharge extreme. Second, for such grid cells, we extracted the maximum annual flooding depth, while the flooding depth of other grid cells was set to zero. The files were then downscaled to a 0.005° resolution using routines implemented in CaMa-Flood[33] (see model description). Population exposure to river floods was assessed by overlaying the population and flooding-depth datasets. When flooding water was present in a 0.005° cell, the population within that cell was considered exposed to flooding.

We accounted for population growth in a separate analysis using population projections from 2006 to 2099 based on shared socioeconomic pathways (SSPs) 1 to 5 provided in the ISIMIP2b framework. The time-varying population datasets were first downscaled to a 0.005° resolution. Population exposure to flooding was then determined using the procedure described above.

**Catchment selection**. Catchments were selected by ensuring that downstream areas were wide, densely populated, and contained major dams. More specifically, the following criteria were used: at least 10 grid cells below dams, a population of at least 5 million residing on the entire main river channel, and the capacity of dams divided by their annual inflow averaged over the number of dams present on the main river channel had to be higher than 0.1. While 15 catchments initially fulfilled these criteria, the Nile catchment was removed from our analysis since a significant portion of its upper section falls within the Köppen–Geiger region BWh (Supplementary Fig. 4), which was (partially) screened out of the analysis. The locations of the remaining 14 catchments are given in Supplementary Fig. 6.

**Catchment flood analysis**. The analysis consisted of two parts: identifying in which grid cells a flood occurred and extracting the corresponding flooded area for those cells. First, daily discharge, collected annually for the 2070–2099 period, in all grid-cells composing the catchments was converted to annual extreme discharges (considering two indices) and compared to the 100-year return extreme discharge. When the annual extreme discharge was higher than the historical 100-year return discharge, a flood was considered to occur in that year. Second, for grid cells where a flood occurred, the maximum flooded area of the grid cell was collected. Finally, we presented the aggregated sum of flood occurrence and flooded area of grid-cells located downstream of dams.

## Data availability
The H08 model is open source and its source code is available online (http://h08.nies.go.jp/h08/index.html). The source code of the CaMa-Flood model can be requested from D. Y. All input data are available through the ISIMIP2b protocol which is freely accessible (https://www.isimip.org/). Detail explanations regarding the coupling procedure, including the new variables introduced in the model and the source file to edit, are available online (https://zenodo.org/record/3701166).

## Code availability
Computer code used for analysis and graphic preparation is available online with explanation (https://zenodo.org/record/3701166).

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

## Acknowledgements

This work was mainly supported by Environment Research and Technology Development Fund (2RF-1802) of the Environmental Restoration and Conservation Agency (grant number JPMEERF20182R02), Japan. It was partially supported by the Japan Society for the Promotion of Science (JSPS) KAKENHI grant number 16H06291. Y.P. acknowledges the support from the National Science Foundation (CAREER Award, grant number 1752729).

## Author contributions

J.B. carried out the simulation and analysis. J.B., N.H., D.Y., and Y.P. commented on and edited the manuscript.

## Competing interests

The authors declare no competing interests.
