## [Peer Review File · Nature Communications]

REVIEWER COMMENTS

Reviewer #1 (Remarks to the Author):

Overall, this is an interesting and well-written paper that provides useful estimates of the impact of dams on climate change projections.

The analysis does not account for population growth and behavior (e.g., people moving away from floodplains if the flooding becomes too severe). Would it be possible to use dynamic estimates of population? If not, would it at least be possible to discuss in the paper how your estimates are expected to change if you account for these factors?

On a related note, it might be good to mention that the number of dams is expected to increase during the current century, and that your estimate of the reduction in flood exposure is therefore a low-end estimate. See, for example, <https://link.springer.com/article/10.1007/s00027-014-0377-0>.

Line 16: 1.1 and 4.6 are the first and third quartiles, but what does the range represent? I guess the uncertainty, but this should be explicitly mentioned.

Line 19: "flood risk" = the number of exposed people? Unclear from the text.

Line 23: "the occurrence of flood events largely decreases," compared to historical conditions or to projections that do not account for water regulation? I'm guessing the latter, but this is not clear from the text.

Line 39: and difficulty of parameterizing reservoir outflows.

Line 52: "states-of-the-art": remove second "s"

Line 72: Nice that the model performs well at the monthly time scale, but the daily time scale is more relevant for floods. How does the model perform at the daily time scale?

Line 81: "flood" needs an "s" at the end (plural)

Line 84: "Result" needs an "s" at the end (plural)

Line 86: Still not sure where the uncertainty range comes from. Different GCMs? By the way, that's a huge uncertainty range. Can this uncertainty range be constrained in some way, e.g., by rejecting GCMs with clearly implausible results?

Line 94: Ah, we are finally told what the uncertainty ranges represent :)

Line 104: I don't see much blue in the figure, but this could be due to the poor quality of the figure (please improve it). If there is very little blue, is it necessary to explain all of this in the main paper instead of the supplement?

Line 157: I think the daily time scale (as opposed to sub-daily) might be sufficient to capture some of the effect of the expected increase in precipitation intensities, so I wouldn't say it "is not considered in the present study."

Figure 2: Not sure why such a weird colorbar shape and colormap scale have been used. A normal rectangular colorbar would have been fine, I think, with a perceptually uniform (!) diverging colormap.

Table 2: This table hurts my eyes. Can you turn this into a boxplot figure?

Reviewer #2 (Remarks to the Author):

Dear Authors, dear Editors,

Many thanks for the opportunity to review this interesting paper and I apologize for the delay in my review. First of all, I want to congratulate the authors for pulling together an interesting, innovative and relevant global analysis of the role of dams in reducing flood exposure under climate change. The paper is overall well-written, has clear figures and follows a logical analysis. The role of dams in flood risk control – or the worsening of floods due to mismanagement – is indeed of critical importance and the understanding is still limited.

However, I do have major remarks on the current scope of the paper. Most of all, the authors present ambitious conclusions on the effect of dams on population flood exposure using only their own model as evidence. The paper does not discuss any empirical evidence, does not validate the model results with actual dams or floods, and does therefore in my viewpoint not provide enough evidence to support the conclusions. The validation of the hydrological model in previous studies alone is not sufficient to draw conclusions on the role of dams in a high impact Nature-branded paper. Similarly, the paper does not refer to other studies investigating the role of dams in flood mitigation and how the results presented here compare.

Given the importance of the topic, I would recommend a revision of the paper which I will be happy to review.

Further to the comments:

- Validation of role of dams: throughout the paper, the authors draw real-world conclusions on the role of dams in flood control based solely on the model outcomes, without any sort of validation, comparison with real-world flood events or assessment of actual evidence of dam water release quantities for flood control. Based on the model alone, the authors make claims such as “Explicitly considering dams in flood climate impact studies significantly offset the population exposed to river flood” (line 145). The authors have not attempted validation or real-world comparison of their modeled results with flood risk in dammed basins. The validation section in the Methods exclusively discusses the validation of the original hydrological models but makes no mention of the validation of peak flows before/after dams; the validation of the basin capacity assumed; comparison of the post-dam flood peaks with reported flood events in any of the world’s dammed basins; comparison of the assumed operational regime with reality; etc. I highly recommend that the authors add these steps of validation to produce convincing outcomes. A good starting point would be to analyze historical flood peaks in the river basins in Table S1 with actual flood events in those basins.
- Narrative on role of dams in floods: The authors describe the role of dams in flood control purely as an efficient and beneficial process. In reality, dams cause huge upstream inundation as well as lethal downstream flooding. Dams are generally operated with decisions that optimize the electricity generation rather than reduce flooding. As such, dams often suddenly release large quantities of water that without warning kill people downstream. Whereas these flood areas may for a given modeled return period be smaller than uncontrolled flood sizes, the impacts can be far larger since the flow is more sudden and unexpected. From a modelling perspective, one could say that a 1/1000 flood can be generated after having several days of medium intense rainfall but bad management procedures. In addition, the construction phase leads to widespread inundation for the creation of the dam lake and involuntary resettlements. Even though the authors will want to focus on the model results exclusively, these aspects should at least be brought into the narrative, with reference to appropriate literature, to prevent a singular message of “dams reduce flood risk by xx%”.
- Integration in literature: The authors claim in several instances that no previous studies have integrated the role of dams in flood risk analysis on a global scale. Whereas this is probably true, the authors should more carefully describe key literature that has attempted this feat on regional and national levels. This includes the new work of Zhao et al (2020) and related papers, which should be brought into the discussion and comparison of results.
- Recommendations: it would be interesting to extend the recommendations regarding (1) the use of the results for flood management, dam operation and early warning; and (2) future research that is needed to

address the limitations of this approach.

Reference mentioned

Zhao et al., 2020. The Impact of Dams on Design Floods in the Conterminous US. Water Resources Research
<https://doi.org/10.1029/2019WR025380>

REVIEWER COMMENTS

Reviewer #1

(Remarks to the Author):

Overall, this is an interesting and well-written paper that provides useful estimates of the impact of dams on climate change projections.

Response: Thank you very much for this positive assessment.

Comment #1: *The analysis does not account for population growth and behavior (e.g., people moving away from floodplains if the flooding becomes too severe). Would it be possible to use dynamic estimates of population? If not, would it at least be possible to discuss in the paper how your estimates are expected to change if you account for these factors?*

Response: Thank you for the important recommendations on how to realistically project global population exposure to future flood risk. Our comments first address the inclusion of population growth in our analysis. We then discuss the consideration of population behaviour in flood analyses.

We accounted for population growth by using population projections based on the Shared Socioeconomic Pathways (SSPs). Using a procedure identical to the one described in the main analysis (Methods, Population exposure section), we downscaled the population projections associated with SSP1 to those in SSP5 (original global data provided in the Inter-Sectoral Impact Model Intercomparison Project, phase 2b at a $0.5^\circ \times 0.5^\circ$ spatial resolution and a yearly temporal resolution). We then assessed future population exposure (all five SSPs) to a 100-year flood with and without dams (P7, L289–292). Figures analogous to the main figure (Fig. 1) were created for all five SSP scenarios and are provided as supplementary figures (Supplementary Figures S10 and S11).

According to these new results, by the end of the 21st century, dams will decrease the number of people exposed historically 100-year flood by 2.2, 2.7, 3.6, 2.8, and 2.2 million in the RCP2.6 scenario and five SSPs, respectively, compared to the no-dam scenario. In the RCP6.0 scenario, the number of people exposed to a historical 100-year flood further decreased by 2.8, 2.9, 8.7, 2.9, and 2.3 million for SSP1 through SSP5, respectively, due to the effect of dams. In the main text, we have briefly mentioned how future population exposure to a 100-year river flood changes depending on the SSP projection (P3 L109–112).

To account for the effect of population behaviour on our global flood analysis (i.e. migration from flood-prone areas), global gridded population datasets depicting these effects are required. As far as we surveyed, the only published analysis containing this feature considers only 18 countries, spread across Africa, Asia, and Latin America¹. In the latest global population distribution projection by Vollset et al.² although economic and geopolitical effects were ambitiously accounted for, the effects of floods were neither considered nor mentioned. To forecast global population density realistically by accounting for human behaviour, the influence of climate change on total risks must be considered. Such risks include health issues (e.g. heat stress, malaria), water resources, typhoons, sea level rise, and wildfires, among many others. Consequently, the creation of such a dataset is outside the scope of this paper. By not accounting for the effects of risk-hedging migration, our estimates of future populations (using the SSP projections described above) exposed to a 100-year flood are likely overestimated in flood-prone areas. We have noted in the conclusion the importance of creating future population distributions that account for population behaviour (P4, L196–201).

Comment #2: *On a related note, it might be good to mention that the number of dams is expected to increase during the current century, and that your estimate of the reduction in flood exposure is therefore a low-end estimate. See, for example, <https://link.springer.com/article/10.1007/s00027-014-0377-0>.*

Response: That the number of dams is expected to increase in this century was briefly stated in the

manuscript (P1, L33–34). However, we did not mention that our estimate of the reduction in flood exposure was on the low end. We have added a sentence at the end of the manuscript (P4, L161-163) stressing this fact. Please note that the suggested article was already cited in the manuscript.

Comment #3: *Line 16: 1.1 and 4.6 are the first and third quartiles, but what does the range represent? I guess the uncertainty, but this should be explicitly mentioned.*

Response: You are correct, the range represents the uncertainty across all GCMs. We have clarified this point accordingly, when the first uncertainty range is introduced (P2, L99–100). Note that the abstract was completely re-written to better convey our main findings.

Comment #4: *Line 19: "flood risk" = the number of exposed people? Unclear from the text.*

Response: Yes, that is correct. We have modified the manuscript to make this clearer (P1, L25-26 and also see P2, L58).

Comment #5: *Line 23: "the occurrence of flood events largely decreases," compared to historical conditions or to projections that do not account for water regulation? I'm guessing the latter, but this is not clear from the text.*

Response: Compared to future projections with no dam implementation. The abstract was re-written to convey our findings more simply (P1, L20-22).

Comment #6: *Line 39: and difficulty of parameterizing reservoir outflows.*

Response: Thank you for this important suggestion. We have added it to the manuscript (P1, L36).

Comment #7: *Line 52: "states-of-the-art": remove second "s"*

Response: Thank you. We have made this correction (P2, L55).

Comment #8: *Line 72: Nice that the model performs well at the monthly time scale, but the daily time scale is more relevant for floods. How does the model perform at the daily time scale?*

Response: Thank you for the valuable comment. The daily time scale is a reasonable time scale for floods.

The two global models, H08 and CaMa-Flood, coupled in this research have been well validated in previous global and regional studies^{3–6}. In short, H08 has been validated mostly at a monthly interval (and at a daily interval in regional applications). Similarly, CaMa-Flood has been validated at a monthly interval at the global scale and at a daily interval in some regional analyses. In the next section, we briefly review the performance of H08 and CaMa-Flood in these earlier applications. A second section discusses some caveats regarding daily validation in our study, and the third section describes what we have newly added to the manuscript.

On a global scale, the H08 model has been intensively validated in its model description papers^{5,7,8}. The model was validated mostly at a monthly interval, because of data availability. The key characteristic of H08 is that it incorporates various human water-management components, including reservoir operation and water abstraction. The model should therefore be validated in heavily developed basins in a wide range of climatic zones using long-term river discharge below dams, from major cities to vast irrigated cropland. Unfortunately, the availability of such data is quite limited and daily data are rarely available in developing countries. Such limitations are not specific to H08 and are commonly seen in other global hydrological models (GHMs). Note that observations of daily river discharge are available at thousands of stations globally (although the global distribution is, again, extremely uneven). When these observations are adjusted to remove and mitigate geographical bias and/or the influence of human management, the GHMs can be validated at a daily interval for many stations (i.e. a performance score can be calculated). However, we believe that such analysis, without considering the effect of dams, would not address your comment. In addition to the validation depicted in the description papers, H08 was intensively compared against the latest GHMs and was reported to perform relatively well (i.e. H08 is seldom in the lower quartile among models for numerous metrics)^{9–13}. The study by Zaherpour et al.⁹

highlighted that the H08 model was among the top three GHMs in term of accurately reproducing the magnitude of flows associated with different return periods.

The H08 model has been applied to regional studies and validated at a daily interval owing to the generous provision of data by local collaborators. In contrast with global-scale studies, the parameters of the model are tuned to better represent local water-management practices and the physical characteristics of watersheds. Consequently, the H08 model has been validated at a daily interval in the Chao-Phraya basin⁶, the Ganges–Brahmaputra–Meghna basin³, and Kyushu Island⁴, and it performed as well as specialized regional hydrological models.

The CaMa-Flood model has also been extensively used and validated^{14–17}. Papers on the model description focus on regional applications, where the model faithfully reproduced daily measurements of river gauging stations across the globe^{14,15}. Of particular interest is the ability of the CaMa-Flood model to reproduce historical flood patterns accurately due to the integration of satellite-based topography data^{14,17}. At a global scale, the sophisticated routing scheme implemented in the CaMa-Flood model has been shown to improve the reproduction of maximum discharge in managed and near-natural basins (in two-thirds of the studied basins) compared to simulations by typical GHMs¹¹. Another recent global study¹⁸ reaffirmed that the discharge simulations produced by the CaMa-Flood were generally reasonable (depending on the forcing data), resulting in simulated flood fractions that correlated highly with observations.

Next, we note that daily validation is not possible in this study because of the input data characteristics. The input data used were developed to analyse the impacts of century-scale climate transitions¹⁹. For this purpose, continuous GCM simulation runs were first prepared for the historical (1860–2005) and future (2006–2100) periods. Because the climate model runs for the historical period were not assimilated using past observed atmospheric and oceanic data, the timing of weather events was not replicated due to the chaotic nature of the climate system. In other words, the input data we used in this study are considered reliable in terms of long-term climate-change transitions but are compatible with reproducing historical daily hydrometeorological events.

Focusing on basins larger than 500,000 km², we extracted the historically reported²⁰ maximum peak flows, which we then compared with simulated maximum discharge during the historical period (from 1861 to 2005). Please note that historical climate data do not reproduce the exact timing of weather events since GCMs were not assimilated, hence we pragmatically assessed to what extent the simulated maximum streamflow was constrained by historical maximum records. In contrast with a previous report²¹, we excluded locations where maximum discharges were the result of ice jams, and for which there was no disclosed information regarding the methodology used to acquire maximum streamflow data. This resulted in a total of 33 locations. The results are presented in Fig. S24 and discussed in a new section in the Supplementary Material (Section S1).

The simulated maximum daily flows were consistent within an order of magnitude with the reported maximum daily flows, indicating that the simulated peak streamflows were plausible, regardless of the use of GCM-based climate forcing. More critically, 83.3% of the simulated maximum streamflows were within $\pm 50\%$ of the observed maximum, implying that the simulated peak streamflows were generally within the precision accuracy associated with the technique used to evaluate historical maximum streamflows.

Note that while the $\pm 50\%$ criterion may seem high, the uncertainty associated with historically unprecedented floods is enormous. Uncertainty associated with the observations was not systematically reported but was more than $\pm 50\%$ in several locations. The USGS reported similar disparities in maximum streamflows, depending on the methodology applied (differences in maximum streamflows ranged from 34% to 74% across basins in the US)²².

We also cited additional validation studies at global and regional scales and highlighted the results that are especially relevant to this study (P2 L74–96).

Comment #9: *Line 81: "flood" needs an "s" at the end (plural)*

Response: Thank you, this has been corrected in the revised manuscript.

Comment #10: *Line 84: "Result" needs an "s" at the end (plural)*

Response: Thank you, this has been corrected in the revised manuscript.

Comment #11: *Line 86: Still not sure where the uncertainty range comes from. Different GCMs? By the way, that's a huge uncertainty range. Can this uncertainty range be constrained in some way, e.g., by rejecting GCMs with clearly implausible results?*

Response: Yes, the range represents the uncertainty from the GCM ensemble. This critical information has been added when the first uncertainty range is given (P2, L99–100).

Constraining the range or rejecting some outliers is difficult since the GCM ensemble is rather small (4). All GCMs are part of the ISIMIP2b protocol and have been used in many global and regional studies^{19,23–25} in which there was no mention of GCM outliers for any of the input data (also see Supplementary Figs. S8, showing that the discharge produced by all GCMs is reasonable compared to observations).

Comment #12: *Line 94: Ah, we are finally told what the uncertainty ranges represent :)*

Response: Yes, this explanation should have been given when the first uncertainty range was reported. As mentioned in the previous comment, we have added this information to earlier sections.

Comment #13: *Line 104: I don't see much blue in the figure, but this could be due to the poor quality of the figure (please improve it). If there is very little blue, is it necessary to explain all of this in the main paper instead of the supplement?*

Response: The blue grid cells aggregate to $6.7 \pm 2.4\%$ and $4.6 \pm 1.1\%$ of the global land surface (excluding Köppen-Geiger regions EF and BWh) for RCP2.6 and RCP6.0, respectively (see P3 L130–131). These grid cells primarily appear in Australia and the central US (see Fig. 1 and Fig. S5c). We have explained how the generic dam release algorithm can produce these results and listed some publications that have highlighted a similar pattern (P3 L128–133). Since these percentages are not negligible and may confuse readers, we would like to keep a short explanation in the main paper.

We have nevertheless improved the main figure by changing the colour palette, the colour bar, and the background colour of the map to improve readability. Supplementary figure S5c has also been modified to match the new map design.

Comment #14: *Line 157: I think the daily time scale (as opposed to sub-daily) might be sufficient to capture some of the effect of the expected increase in precipitation intensities, so I wouldn't say it "is not considered in the present study."*

Response: Thank you for the comment. We have modified the sentence to read “not fully considered” (P4, L177).

Comment #15: *Figure 2: Not sure why such a weird colorbar shape and colormap scale have been used. A normal rectangular colorbar would have been fine, I think, with a perceptually uniform (!) diverging colormap.*

Response: Our intention was to provide information about the proportion of each category in addition to their locations on the map. We have changed the colormap scale to a classical diverging palette. We have also changed the colorbar to a traditional histogram located on the right side of the global map. Supplementary Fig. S5c has also been modified to match the new aesthetic.

Comment #16: *Table 2: This table hurts my eyes. Can you turn this into a boxplot figure?*

Response: We assume you meant Table 1? The table has been changed to a boxplot figure, as suggested,

and is now labelled Fig. 3. The manuscript has been edited to include the new figure number.
Supplementary Table S1 has also been modified to a boxplot for consistency and is now labelled Fig. S7.

Reviewer #2

(Remarks to the Author):

Dear Authors, dear Editors,

Many thanks for the opportunity to review this interesting paper and I apologize for the delay in my review. First of all, I want to congratulate the authors for pulling together an interesting, innovative and relevant global analysis of the role of dams in reducing flood exposure under climate change. The paper is overall well-written, has clear figures and follows a logical analysis. The role of dams in flood risk control – or the worsening of floods due to mismanagement – is indeed of critical importance and the understanding is still limited.

However, I do have major remarks on the current scope of the paper. Most of all, the authors present ambitious conclusions on the effect of dams on population flood exposure using only their own model as evidence. The paper does not discuss any empirical evidence, does not validate the model results with actual dams or floods, and does therefore in my viewpoint not provide enough evidence to support the conclusions. The validation of the hydrological model in previous studies alone is not sufficient to draw conclusions on the role of dams in a high impact Nature-branded paper. Similarly, the paper does not refer to other studies investigating the role of dams in flood mitigation and how the results presented here compare.

Given the importance of the topic, I would recommend a revision of the paper which I will be happy to review.

Response: We are thankful for the constructive yet critical remarks.

Further to the comments:

Comment: *Validation of role of dams: throughout the paper, the authors draw real-world conclusions on the role of dams in flood control based solely on the model outcomes, without any sort of validation, comparison with real-world flood events or assessment of actual evidence of dam water release quantities for flood control. Based on the model alone, the authors make claims such as “Explicitly considering dams in flood climate impact studies significantly offset the population exposed to river flood” (line 145). The authors have not attempted validation or real-world comparison of their modeled results with flood risk in dammed basins. The validation section in the Methods exclusively discusses the validation of the original hydrological models but makes no mention of the validation of peak flows before/after dams; the validation of the basin capacity assumed; comparison of the post-dam flood peaks with reported flood events in any of the world’s dammed basins; comparison of the assumed operational regime with reality; etc. I highly recommend that the authors add these steps of validation to produce convincing outcomes. A good starting point would be to analyze historical flood peaks in the river basins in Table S1 with actual flood events in those basins.*

Response #17: Thank you for bringing the crucial point of validation to our attention. Our answers to fully address your concerns cover validation of dam reservoir operation (peak flow pre/post dam), and assessment of basin capacity.

We assessed the validity of dam reservoir operations at 11 locations by comparing monthly peak flow before and after the presence of a dam using publicly available and personally communicated streamflow data.

- (i) We used a daily observation dataset for Nakhon Sawan (Thailand) that was used in a previous analysis by some of the authors. The dataset covers the period from 1956 to 2010. In 1964 and 1974, two major dams, the Bhumibol Dam and the Sirikit Dam, officially began operation. We therefore divided the daily flow into pre- and post-dam categories (before 1964 and after 1974, discarding the period between 1964 and 1974), matching our model

experimental setup, and compared monthly average flows for dam and no-dam conditions to those of our simulations. See Supplementary Figure S13.

- (ii) We also leveraged the study by Mei et al.²⁶ in which the authors investigated the effects of dams on flood occurrence in the US. We limited our analysis to large dams in the Grand database with catchment areas are larger than 2,000 km², resulting in 10 locations (see Supplementary Table 3). Streamflow observations were retrieved from the USGS website and were separated into two periods: before and after dam construction. We then overlaid our simulations using the appropriate experiment (with or without dam implementation). All figures are available in the Supplementary Material (Figs. S14 through S23).

After dam construction, low streamflow increased while high streamflow decreased as a result of water storage and controlled release. Overall, monthly fluctuations in streamflow were greatly attenuated. This is in line with observations and highlights that dam operations are well captured by the generic release algorithm employed in our study. Considerable inter-annual variations are shown by both observations and simulations, reflecting inter-annual fluctuations in precipitation. Note that, due to the absence of data assimilation in the GCMs, a comparison of year-to-year variability (a hydrograph) is not meaningful for this study.

Here, we interpret the term basin capacity in two ways. First, we take it as ‘channel capacity’, which is directly relevant to the occurrence and extent of flooding. Second, we explain it as ‘basin water storage capacity,’ which is directly relevant to the hydrological response.

Regarding the first interpretation, the capacity of a river channel to hold water during a flood is directly related to the width and bank height parameters of the river channel. In CaMa-Flood v3.6, a satellite product called GWD-LR is used to estimate river width¹⁵ (the main feature of this product is that it accounts for islands within rivers). The effective river width in GWD-LR has been reported to be slightly narrower compared to other databases (with a relative difference of $\pm 20\%$ for most river channels). In contrast, bank height is empirically determined using annual river discharge¹⁶. The sensitivity of these two parameters for predicting river streamflow and floodplain inundation was investigated by Yamazaki et al.¹⁶. Flooding from river channels happens more frequently in the case of relatively low bank heights and narrow channels.

As for the second interpretation, the most straightforward approach is to compare simulated terrestrial water storage (TWS) with satellite observations by the Gravity Recovery and Climate Experiment (GRACE). Comparison under exact simulation conditions (i.e. ISIMIP2b framework) has been systematically conducted by one of the authors (Yadu Pokhrel) with a paper under review. Briefly, the seasonal march of terrestrial water storage is well reproduced in the H08 model as well as in the other six most recent and advanced global hydrological models. We will refer to this paper as soon as it is accepted. Note that a detailed comparison of simulated and observed TWS for 12 major basins is presented in the H08 model description paper⁵. It shows good agreement between observations and simulations for both developed (e.g. the Mississippi River) and underdeveloped (e.g. the Congo River) basins.

Last, one challenge in the validation of real-world flood data within our framework is that, while the GCMs employed in this analysis replicate past (historical) climate, they do not reproduce the exact timing of weather events since the GCMs were not assimilated. This impeded the traditional comparison of observed and simulated hydrographs, which was not done, due to: GCM projections that were not assimilated, limited availability of daily streamflow records, and limited availability of water-management data (e.g. reservoir operation, water withdrawal) to investigate daily reproducibility.

In summary, the main text of the manuscript has been amended by:

- (i) Adding details about the calibration of both models at a daily time scale (P2, L74–88).
- (ii) Adding the monthly validation of discharge, both pre- and post-dams (P2, L88–91).

- (iii) Highlighting that the patterns and range of future global floods are similar to those presented in a previous publication (P2 L93–94).
- (iv) Clearly discussing the comparison of forecasted population exposure to a 100-year flood with one or more public databases (P2 L94–96).
- (v) Adding information about the performance of the H08 model versus other global hydrological models together with its performance in reproducing streamflows associated with different return periods (P2, L76–78).

The Supplementary Material has been amended by:

- (i) Creating of Figures S13 through Fig. S23 to show the validation of monthly discharges before and after dam implementation.
- (ii) Adding Fig. S24 to show observed and simulated maximum daily streamflows for 33 major catchments.
- (iii) Adding a supplementary section (Section S1).

Comment #18: *Narrative on role of dams in floods: The authors describe the role of dams in flood control purely as an efficient and beneficial process. In reality, dams cause huge upstream inundation as well as lethal downstream flooding. Dams are generally operated with decisions that optimize the electricity generation rather than reduce flooding. As such, dams often suddenly release large quantities of water that without warning kill people downstream. Whereas these flood areas may for a given modeled return period be smaller than uncontrolled flood sizes, the impacts can be far larger since the flow is more sudden and unexpected. From a modelling perspective, one could say that a 1/1000 flood can be generated after having several days of medium intense rainfall but bad management procedures. In addition, the construction phase leads to widespread inundation for the creation of the dam lake and involuntary resettlements. Even though the authors will want to focus on the model results exclusively, these aspects should at least be brought into the narrative, with reference to appropriate literature, to prevent a singular message of “dams reduce flood risk by xx%”.*

Response: Thank you for this important suggestion. We note that your points are to acknowledge the negative effects of dams on flooding, including the consequences of mismanagement; the adverse social and ecological consequences of dam construction; and how these point colour the conclusions shown we drew in the original manuscript.

When properly operated, dams are reported to mitigate downstream floods²⁷. For example, in 2011, there was a massive flood in Central Thailand, despite two major dams (Bhumibol and Sirikit) that collectively stored about 10,000,000,000 m³ of water (see Komori et al.²⁸). Similarly, in Japan, two dams (the Miyagase and Shiroyama dams) that collectively stored 72,000,000 million m³ of water were reported to prevent an additional 1.1 m of flood depth downstream during typhoon Hagibis in October 2019 (see https://www.ktr.mlit.go.jp/ktr_content/content/000760679.pdf; report produced by the Ministry of Land, Infrastructure, Transport and Tourism (Japan); in Japanese). Although modifying existing reservoir operations can provide additional benefits with regards to flood risk (e.g. by lowering flooding depth and flooded area)⁶, prioritizing water security is often necessary to offset the effects of climate change, potentially at the cost of more effective flood mitigation²⁹.

Mismanaged dams also increase the risk of floods, and unexpected accidents may cause fatal disasters downstream. Approximately 14,000 dams in the US are classified as “high hazard potential,” indicating that misoperation would likely result in loss of life^{30,31}. Between 2008 and 2011, improper dam release in India resulted in approximately 600 deaths and the displacement of 309,250 people. More generally, many fatalities have been caused by the release of water from dams without relevant downstream information³².

Globally, the failure of 70 large dams (> 15 m in height) due to flooding has been reported. Among those, 15 failures caused a total of over 10,000 fatalities³². Although many dams built in the US before 1930 failed with few or no fatalities, dam failures between 1970 and 1980 resulted in about 500 deaths

and \$2 billion in losses and damage³³. Dam failure was listed in the Dartmouth database as the main cause of flood in nine events in the US, Australia, and Africa, resulting in the displacement of roughly 9,800 people.

As you pointed out, the construction and filling phases of dams are not considered in our framework. While failure during construction has been reported for over 10 dams (higher than 50 m), this aspect has been generally overlooked³². These phases have, to our knowledge, never been modelled, even in the recently developed reservoir operation schemes which instead focus on capturing the interactions of, and coordination between multiple reservoirs,^{34,35} and balancing the need of human ecosystems³⁶.

The conclusion of the original manuscript described dams as being purely efficient and beneficial. We have amended the manuscript to prevent such a singular message, stressing some of the negative environmental and societal aspects of dams. We have also clearly indicated that we emphasized dams as beneficial given our investigation of the role dams play in reducing global flood exposure under conditions of climate change (P4 L187-189).

To summarize, the manuscript has been amended by:

- 1) Adding a caveat about the construction and filling phases of dams and indicating that accident/mismanagement is not accounted for in the current modelling framework (P4, L178–181)
- 2) Stressing in the conclusion that focus of the role of dams in reducing global flood exposure under climate change generally paints them as beneficial, despite their many negative environmental and social aspects, which must be thoughtfully evaluated alongside potential benefits for the future sustainable development of water resources (P4, L187–196)
- 3) Adding a supplementary section (Section S2) with appropriate references to discuss the further role of dams in floods.

Comment #19: *Integration in literature: The authors claim in several instances that no previous studies have integrated the role of dams in flood risk analysis on a global scale. Whereas this is probably true, the authors should more carefully describe key literature that has attempted this feat on regional and national levels. This includes the new work of Zhao et al (2020) and related papers, which should be brought into the discussion and comparison of results.*

Response: Thank you for bringing this to our attention. We have carefully reviewed the you referenced as well as other relevant literature^{26,37,38}.

The manuscript was edited to include key results obtained of previous studies assessing flood hazard while implementing dam or/and flood defence in a statistical framework (to include the Zhao et al. study³⁸) (P1,2 L48-52 and P3 L117–118).

We have also compared key results with the literature wherever possible (P3, L118–120) in the main manuscript and have added a supplementary section (Section S3) to discuss some results further.

Comment #20: *Recommendations: it would be interesting to extend the recommendations regarding (1) the use of the results for flood management, dam operation and early warning; and (2) future research that is needed to address the limitations of this approach.*

Response: Thank you for the suggestions. We first discuss how the results may apply to various communities and then address possible area for future research.

The manuscript conclusion has been edited to better state the following (P4, L183–196):

- Most studies^{39–41} on climate change risk assessment ignored the effects of river infrastructure. Globally, dams reduced the number of people exposed to 100-year floods by, an average of 14.5%. This number however varies substantially across basins.

- Given that, for most of the world, climate change considerably increases the risk of floods, new reservoir operations will be needed to retain the levels of flood protection that dams have provided in the past.
- Precise, reliable hydro-meteorological forecast will be necessary to maximize the flood-protection capability of dams by limiting untimely and/or excessive outflow.

The manuscript has been similarly edited (P4, L191–196) to address future research needs. Specifically, we have mentioned the need to address existing disparities and uncertainties in global scale datasets (such as location of dams and river networks); develop realistic future population scenarios by accounting for population behaviour; enhance existing historical GCM simulations by assimilating past observations; and (iv) increase the global availability of reservoir operation, daily discharge, and inundation maps to allow for robust model validation.

Cited references:

1. Smith, A. *et al.* New estimates of flood exposure in developing countries using high-resolution population data. *Nature Communications* **10**, 1814 (2019).
2. Vollset, S. E. *et al.* Fertility, mortality, migration, and population scenarios for 195 countries and territories from 2017 to 2100: a forecasting analysis for the Global Burden of Disease Study. *The Lancet* doi:10.1016/S0140-6736(20)30677-2.
3. Masood, M., Yeh, P. J.-F., Hanasaki, N. & Takeuchi, K. Model study of the impacts of future climate change on the hydrology of Ganges–Brahmaputra–Meghna basin. *Hydrology and Earth System Sciences* **19**, 747–770 (2015).
4. HANASAKI, N., FUJIWARA, M., MAJI, A. & SETO, S. ON THE APPLICABILITY OF THE H08 GLOBAL WATER RESOURCES MODEL TO THE KYUSYU ISLAND. *Journal of Japan Society of Civil Engineers, Ser. B1 (Hydraulic Engineering)* **74**, I_109-I_114 (2018).
5. Hanasaki, N., Yoshikawa, S., Pokhrel, Y. & Kanae, S. A global hydrological simulation to specify the sources of water used by humans. *Hydrology and Earth System Sciences* **22**, 789–817 (2018).
6. Mateo, C. M. *et al.* Assessing the impacts of reservoir operation to floodplain inundation by combining hydrological, reservoir management, and hydrodynamic models. *Water Resources Research* **50**, 7245–7266 (2014).
7. Hanasaki, N., Kanae, S. & Oki, T. A reservoir operation scheme for global river routing models. *Journal of Hydrology* **327**, 22–41 (2006).
8. Hanasaki, N., Inuzuka, T., Kanae, S. & Oki, T. An estimation of global virtual water flow and sources of water withdrawal for major crops and livestock products using a global hydrological model. *Journal of Hydrology* **384**, 232–244 (2010).
9. Zaherpour, J. *et al.* Worldwide evaluation of mean and extreme runoff from six global-scale hydrological models that account for human impacts. *Environmental Research Letters* **13**, 065015 (2018).
10. Hattermann, F. F. *et al.* Cross-scale intercomparison of climate change impacts simulated by regional and global hydrological models in eleven large river basins. *Climatic Change* **141**, 561–576 (2017).
11. Zhao, F. *et al.* The critical role of the routing scheme in simulating peak river discharge in global hydrological models. *Environmental Research Letters* **12**, 075003 (2017).
12. Krysanova, V. *et al.* How the performance of hydrological models relates to credibility of projections under climate change. *Hydrological Sciences Journal* **63**, 696–720 (2018).
13. Krysanova, V. *et al.* Intercomparison of regional-scale hydrological models and climate change impacts projected for 12 large river basins worldwide—a synthesis. *Environmental Research Letters* **12**, 105002 (2017).
14. Yamazaki, D., Sato, T., Kanae, S., Hirabayashi, Y. & Bates, P. D. Regional flood dynamics in a bifurcating mega delta simulated in a global river model. *Geophysical Research Letters* **41**, 3127–3135 (2014).
15. Yamazaki, D. *et al.* Development of the Global Width Database for Large Rivers. *Water Resources Research* **50**, 3467–3480 (2014).
16. Yamazaki, D., Kanae, S., Kim, H. & Oki, T. A physically based description of floodplain inundation dynamics in a global river routing model. *Water Resources Research* **47**, (2011).
17. Yamazaki, D. *et al.* Analysis of the water level dynamics simulated by a global river model: A case study in the Amazon River. *Water Resources Research* **48**, (2012).
18. Wei, Z. *et al.* Identification of uncertainty sources in quasi-global discharge and inundation simulations using satellite-based precipitation products. *Journal of Hydrology* **589**, 125180 (2020).
19. Frieler, K. *et al.* Assessing the impacts of 1.5 degree C global warming – simulation protocol of the Inter-Sectoral Impact Model Intercomparison Project (ISIMIP2b). *Geoscientific Model Development* **10**, 4321–4345 (2017).
20. Rodier, J. A. & Roche, M. *World catalogue of maximum observed floods.* (Wallingford (UK) IAHS, 1984).

21. O'Connor, J., E. & Costa, J., E. The World's Largest Floods, Past and Present: Their Causes and Magnitudes. (2004).
22. Capesius P., J. & Rick L., A. Comparison of Two Methods for Estimating Base Flow in Selected Reaches of the South Platte River, Colorado. (2012).
23. Hong, C. *et al.* Impacts of climate change on future air quality and human health in China. *Proc Natl Acad Sci USA* **116**, 17193 (2019).
24. Samaniego, L. *et al.* Propagation of forcing and model uncertainties on hydrological drought characteristics in a multi-model century-long experiment in large river basins. *Climatic Change* **141**, 435–449 (2017).
25. Mishra, V. *et al.* Multimodel assessment of sensitivity and uncertainty of evapotranspiration and a proxy for available water resources under climate change. *Climatic Change* **141**, 451–465 (2017).
26. Mei, X., Van Gelder, P. H. A. J. M., Dai, Z. & Tang, Z. Impact of dams on flood occurrence of selected rivers in the United States. *Frontiers of Earth Science* **11**, 268–282 (2017).
27. Mueller, E. R. *et al.* Geomorphic change and sediment transport during a small artificial flood in a transformed post-dam delta: The Colorado River delta, United States and Mexico. *Ecological Engineering* **106**, 757–775 (2017).
28. Komori, D. *et al.* Characteristics of the 2011 Chao Phraya River flood in Central Thailand. *Hydrological Research Letters* **6**, 41–46 (2012).
29. Ehsani, N., Vörösmarty, C. J., Fekete, B. M. & Stakhiv, E. Z. Reservoir operations under climate change: Storage capacity options to mitigate risk. *Journal of Hydrology* **555**, 435–446 (2017).
30. FEMA. Identifying High hazard Dam Risk in the United States.
31. Ho, M. *et al.* The future role of dams in the United States of America. *Water Resources Research* **53**, 982–998 (2017).
32. Lempérière, F. Dams and Floods. *Engineering* **3**, 144–149 (2017).
33. May, P., J. & Williams, W. *Disaster Policy Implementation, Managing Programs under Shared Governance*. (Springer, Boston, MA, 1986).
34. Rougé, C. *et al.* Coordination and Control: Limits in Standard Representations of Multi-Reservoir Operations in Hydrological Modeling. *Hydrology and Earth System Sciences Discussions* **2019**, 1–37 (2019).
35. Shin, S., Pokhrel, Y. & Miguez-Macho, G. High-Resolution Modeling of Reservoir Release and Storage Dynamics at the Continental Scale. *Water Resources Research* **55**, 787–810 (2019).
36. Yin, X.-A., Yang, Z.-F. & Petts, G. E. Reservoir operating rules to sustain environmental flows in regulated rivers. *Water Resources Research* **47**, (2011).
37. Assani, A. A., Stichelbout, É., Roy, A. G. & Petit, F. Comparison of impacts of dams on the annual maximum flow characteristics in three regulated hydrologic regimes in Québec (Canada). *Hydrological Processes* **20**, 3485–3501 (2006).
38. Zhao, G., Bates, P. & Neal, J. The Impact of Dams on Design Floods in the Conterminous US. *Water Resources Research* **56**, e2019WR025380 (2020).
39. Hirabayashi, Y. *et al.* Global flood risk under climate change. *Nature Climate Change* **3**, 816 (2013).
40. Jongman, B., Ward, P. J. & Aerts, J. C. J. H. Global exposure to river and coastal flooding: Long term trends and changes. *Global Environmental Change* **22**, 823–835 (2012).
41. Ward, P. J. *et al.* Assessing flood risk at the global scale: model setup, results, and sensitivity. *Environmental Research Letters* **8**, 044019 (2013).

REVIEWERS' COMMENTS

Reviewer #1 (Remarks to the Author):

The authors have done a good job of addressing the comments and I recommend accepting the paper.

Reviewer #2 (Remarks to the Author):

Dear authors

Thank you for the detailed revisions of the manuscript, which address most of my main concerns raised. I am fine with the additional validation provided in the revised manuscript.

My concerns around the messaging remain to some extent, whereas some improvements were made. As reflected in several papers in Nature-branded journals, the flood protection benefits of dams are in many cases outweighed by the dramatic negative consequences on biodiversity and downstream populations, especially in semi-natural basins. It would be great if this could be added with a few words at the end of the abstract, as well as in places like line 161 where it is mentioned that "dam benefits in the Amazon, Congo, and Mekong rivers were relatively small due to the small number of mainstem dams. Nevertheless, hundreds of dams are planned that could offer similar flood benefits in the future." As the authors know, several areas in the United States are currently 'undamming' to reverse the negative impacts although possibly increasing some of the flood risk.

After the messaging is finetuned I am happy with the final manuscript.